# Pre-Trained Language Models Augmented with Synthetic Scanpaths for Natural Language Understanding

**Shuwen Deng[1], Paul Prasse[1], David R. Reich[1], Tobias Scheffer[1], Lena A. Jäger[1,2]**

[1] Department of Computer Science, University of Potsdam, Germany
[2] Department of Computational Linguistics, University of Zurich, Switzerland
{deng, prasse, david.reich, tobias.scheffer}@uni-potsdam.de
jaeger@cl.uzh.ch

## Abstract

Human gaze data offer cognitive information that reflects natural language comprehension. Indeed, augmenting language models with human scanpaths has proven beneficial for a range of NLP tasks, including language understanding. However, the applicability of this approach is hampered because the abundance of text corpora is contrasted by a scarcity of gaze data. Although models for the generation of human-like scanpaths during reading have been developed, the potential of synthetic gaze data across NLP tasks remains largely unexplored. We develop a model that integrates synthetic scanpath generation with a scanpath-augmented language model, eliminating the need for human gaze data. Since the model's error gradient can be propagated throughout all parts of the model, the scanpath generator can be fine-tuned to downstream tasks. We find that the proposed model not only outperforms the underlying language model, but achieves a performance that is comparable to a language model augmented with real human gaze data. Our code is publicly available.[1]

## 1 Introduction and Related Work

When humans read, they naturally engage in the cognitive process of comprehending language, which, in turn, is reflected in their gaze behavior (Just and Carpenter, 1980). In a nutshell, a scanpath (i.e., sequence of consecutive fixations) on a stimulus text approximates the reader's attention, which can be exploited to inform Natural Language Processing (NLP) tasks.

Gaze data has been shown to be beneficial in various NLP tasks, such as part-of-speech-tagging (Barrett et al., 2016), named entity recognition (Hollenstein and Zhang, 2019), generating image captions (Takmaz et al., 2020) and question answering (Sood et al., 2021). Researchers have explored

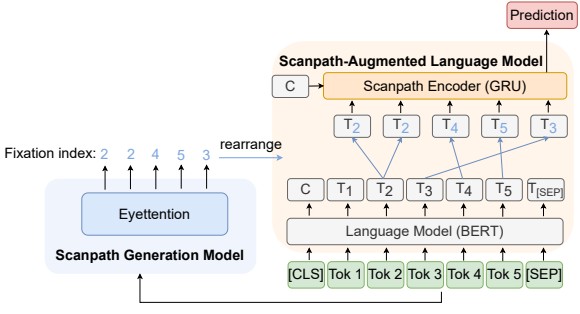

Figure 1: Synthetic scanpath-augmented language model: the Scanpath Generation Model predicts a sequence of fixations for an input sentence; token embeddings are rearranged according to the order of fixations.

the use of aggregated word-level gaze features to regularize neural attention mechanisms (Barrett et al., 2018; Sood et al., 2020). Moreover, non-aggregated scanpaths, which capture the complete sequential ordering of the reader's gaze behavior, have also demonstrated promise in NLP tasks (Mishra et al., 2017, 2018a; Yang and Hollenstein, 2023).

However, collecting gaze data is a resource-intensive endeavor, even for very small text corpora. Hence, human gaze data is scarce, and NLP task-specific gaze recordings are even scarcer. Moreover, applying a language model that additionally consumes gaze data requires gaze data to be available for the input text at deployment time—which is unrealistic for most use cases. To overcome these limitations, researchers have proposed a multi-task learning approach for NLP tasks such as sentence compression (Klerke et al., 2016), sentiment analysis (Mishra et al., 2018b), and predicting text readability (González-Garduño and Søgaard, 2017). In this approach, labeled data for the specific NLP task is used as the primary task, while a separate eye-tracking corpus is utilized as an auxiliary task. While this approach helps mitigate the need for task-specific gaze data during training and testing, the problem of general scarcity of gaze samples

---

[1] https://github.com/aeye-lab/EMNLP-SyntheticScanpaths-NLU-PretrainedLM.

remains and hinders effective supervision for data-intensive architectures.

In this paper, we propose an alternative approach by using synthetic gaze data, which can be generated easily for any given text, to provide cognitive signals across NLP tasks. The seminal work of Sood et al. (2020), which integrates eye movement data generated by a computational cognitive model of eye-movement-control-during-reading for tasks such as sentence compression and paraphrase generation, demonstrated the potential of synthetic eye-gaze data. Khurana et al. (2023) explored a proof-of-concept model that integrated synthetic gaze data across multiple NLP tasks, but their results did not reach the performance of a fine-tuned BERT model (Devlin et al., 2019) without eye gaze on the General Language Understanding Evaluation (GLUE) benchmark. In our work, we build on recent advances in the development of machine-learning models for generating human-like scanpaths during reading (Deng et al., 2023; Bolliger et al., 2023; Khurana et al., 2023; Nilsson and Nivre, 2011).

We develop a model that combines synthetic scanpath generation with a scanpath-augmented language model, eliminating the need for human gaze data. The model allows for fine-tuning the scanpath generator to downstream tasks by propagating the error gradient through the entire model. Our approach not only outperforms the underlying language model in multiple tasks on the GLUE, especially in low-resource settings, but even reaches a performance comparable to an eye-gaze augmented model that uses real, rather than synthetic, eye movement data in sentiment classification.

## 2 Model

We develop a model that combines a scanpath generation model with a scanpath-augmented language model to perform NLP downstream tasks. Figure 1 depicts the proposed model architecture.

**Scanpath Generation Model** We adopt Eyettention (Deng et al., 2023), an open-source state-of-the-art model for scanpath generation over text. Eyettention predicts consecutive fixation locations, represented as word indices, based on a stimulus sentence and the preceding fixations. It consists of two encoders, one for embedding the stimulus sentence, and the other for embedding the scanpath history. A cross-attention layer aligns the outputs of the two encoders, and a decoder produces a

probability distribution over saccade ranges at each timestep. The next fixated word index is determined by sampling from this distribution.

**Scanpath-Augmented Language Model** We adopt the PLM-AS framework (Yang and Hollenstein, 2023), which augments pre-trained language models with human scanpaths for sentiment classification. This framework uses a language model to extract token embeddings for a sentence, associating each embedding with its position index. By utilizing a human scanpath (fixation index sequence) as input, the model rearranges the token embedding sequence based on the order in which the words are fixated by the reader. The transformed sequence is then fed into a scanpath encoder, implemented as a layer of gated recurrent units (GRU), where the output of the last step is used as the final feature for sentiment classification. This framework allows for the use of different language models and achieves high performance through fine-tuning. In this work, we employ $BERT_{BASE}$[2] (Devlin et al., 2019) as the language model, following Yang and Hollenstein (2023).

**Joint Modeling for NLP Tasks** To eliminate the need for human gaze data, we integrate the synthetic scanpath generated by the Eyettention model consisting of a fixation index sequence into the PLM-AS framework. Before integration, the word index sequence generated by Eyettention is converted into a token index sequence. During training, the error gradient of the scanpath-augmented language model can be back-propagated through the Eyettention model, allowing its parameters to be adapted for a specific NLP task. To handle the non-differentiable sampling from a categorical distribution involved in scanpath generation, we employ the Gumbel-softmax distribution (Jang et al., 2017) as a fully differentiable approximation. The training process consists of two phases. First, we pre-train the Eyettention model on a natural reading task. Second, we train the entire model, which includes fine-tuning the language model and the Eyettention model, as well as training the scanpath encoder from scratch. For the Eyettention model, we add residual connections in both encoders to enhance its performance.

---

[2] Note that BERT can be substituted with other advanced pre-trained language models, potentially leading to further enhancements in task performance.

# 3 Experiments

In this section, we describe the data and present the evaluation results of our model for a wide range of NLP tasks. Further details about training and hyperparameter tuning can be found in Appendix B.

## 3.1 Data Sets

**CELER** (Berzak et al., 2022): We pre-train the scanpath generation model Eyettention on the L1 subset of CELER, which contains eye-tracking recordings collected from 69 native speakers of English during natural reading of 5,456 sentences.

**ETSA** (Mishra et al., 2016) contains task-specific gaze recordings for sentiment classification of 7 subjects who each read 383 positive and 611 negative sentences, including sarcastic quotes, short movie reviews, and tweets.

**GLUE** (Wang et al., 2018) includes sentiment analysis (SST-2), linguistic acceptability (CoLA), similarity and paraphrase tasks (MRPC, STS-B, QQP), and natural language inference tasks (MNLI, QNLI, RTE). No gaze data are available.

## 3.2 Sentiment Classification

Table 1 presents the results of our model on the sentiment classification task ETSA (Mishra et al., 2016), in comparison to BERT and previous state-of-the-art eye-gaze augmented models. We follow a 10-fold cross-validation regime. In each iteration, BERT is fine-tuned on the training portion of the ETSA text corpus, and PLM-AS is fine-tuned on the training portion of the ETSA text corpus and gaze data. Our model is fine-tuned on the training portion of the ETSA text corpus and, instead of the ETSA gaze data, synthetic gaze data generated by Eyettention. Since each sentence is associated with multiple scanpaths, we compute the final prediction by averaging the pre-softmax logits obtained from the models across all scanpaths for the PLM-AS baseline. Our model averages equally many synthetic scanpaths. We make multiple notable observations in Table 1:

(a) Our model outperforms both BERT and the state-of-the-art ScanTextGAN (Khurana et al., 2023) augmented with gaze data.

(b) Our model, augmented with *synthetic* scanpaths, achieves comparable performance to the PLM-AS model augmented with *human* scanpaths, eliminating the need for human scanpaths.

(c) Ablation experiments (bottom two rows) show that when the Eyettention model is frozen or

| Model | Scanpath (#) | F1 | AUC |
|---|---|---|---|
| BERT⋆ | - | $82.93_{2.26}$ | $92.42_{1.62}$ |
| ScanTextGAN | real | 83.34 | - |
| ScanTextGAN | synthetic | 84.77 | - |
| PLM-AS⋆ | real (7) | $\mathbf{85.81}_{1.16}$ | $94.79_{1.02}$ |
| Ours⋆ | synthetic (7) | $85.35_{1.77}$ | $\mathbf{94.90}_{0.94}$ |
| Eyettention (frozen)⋆ | synthetic (7) | $84.52_{1.79}$ | $94.50_{1.03}$ |
| Eyettention (scratch)⋆ | synthetic (7) | $85.03_{1.6}$ | $94.77_{1.03}$ |

Table 1: Results for sentiment classification on ETSA, with standard errors indicated as subscript. Results obtained from our experiments are marked with ⋆; other results are from the respective papers for recapitulation.

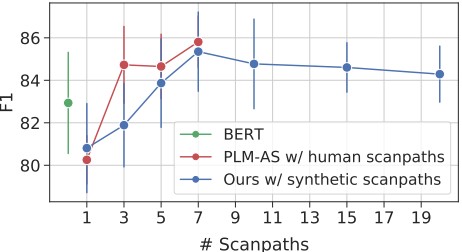

Figure 2: Sentiment classification performance on ETSA with varying numbers of scanpaths at training and application time. Error bars show the standard error.

not pre-trained, the performance decreases. This demonstrates the importance of both pre-training and task-specific fine-tuning of the scanpath generator.

**Varying the number of scanpaths** We analyze the impact of the number of scanpaths sampled both at training and at application time on model performance. Figure 2 shows the F1 score as a function of the number of scanpaths used by BERT without eye gaze, PLM-AS with human scanpaths, and our model with synthetic scanpaths. We observe that the performance of scanpath-augmented models improves as the number of scanpaths increases, reaching its peak at seven scanpaths.[3] Importantly, our model outperforms BERT and, when being augmented with five or more synthetic scanpaths, approaches the performance of PLM-AS augmented with human scanpaths.

**Low-Resource Performance** We hypothesize that eye gaze might be most beneficial in low-resource settings. To test this hypothesis, we sample a small subset of the training sentences $K = \{200, 400, 600\}$ from the total number of around 800 training instances, and evaluate the performance of our model augmented with seven syn-

---

[3]The optimal number of scanpaths to be used by the model is considered a hyperparameter for the subsequent experiments.

| K | Model | Gaze | MNLI 392k | QQP 363k | QNLI 108k | SST-2 67k | CoLA 8.5k | STS-B 5.7k | MRPC 3.5k | RTE 2.5k | Avg. - |
|---|---|---|---|---|---|---|---|---|---|---|---|
| 200 | BERT | ✗ | $42.90_{1.51}$ | $57.42_{2.03}$ | $\mathbf{73.07_{0.16}}$ | $78.78_{1.10}$ | $16.95_{2.74}$ | $\mathbf{79.43_{0.69}}$ | $81.18_{0.04}$ | $54.30_{1.50}$ | 60.50 |
| 200 | Ours | ✓ | $\mathbf{48.97_{0.83}}$ | $\mathbf{61.63_{1.78}}$ | $70.46_{0.62}$ | $\mathbf{80.76_{0.74}}$ | $\mathbf{24.08_{3.55}}$ | $74.94_{1.20}$ | $\mathbf{81.85_{0.17}}$ | $\mathbf{59.35_{1.47}}$ | 62.75 |
| 500 | BERT | ✗ | $52.09_{1.05}$ | $65.13_{0.37}$ | $77.04_{0.19}$ | $82.55_{0.47}$ | $35.61_{1.74}$ | $\mathbf{83.14_{0.41}}$ | $81.53_{0.29}$ | $60.72_{0.61}$ | 67.23 |
| 500 | Ours | ✓ | $\mathbf{56.48_{0.38}}$ | $\mathbf{67.81_{0.23}}$ | $\mathbf{77.60_{0.26}}$ | $\mathbf{84.63_{0.50}}$ | $\mathbf{36.41_{1.39}}$ | $81.99_{0.58}$ | $\mathbf{82.32_{0.52}}$ | $\mathbf{61.88_{1.24}}$ | 68.64 |
| 1000 | BERT | ✗ | $58.97_{0.58}$ | $67.35_{0.49}$ | $78.88_{0.36}$ | $85.80_{0.55}$ | $39.89_{1.64}$ | $\mathbf{85.42_{0.21}}$ | $84.18_{1.00}$ | $63.39_{0.99}$ | 70.49 |
| 1000 | Ours | ✓ | $\mathbf{61.28_{0.25}}$ | $\mathbf{70.65_{0.14}}$ | $\mathbf{80.74_{0.10}}$ | $\mathbf{86.06_{0.29}}$ | $\mathbf{41.19_{0.50}}$ | $85.13_{0.43}$ | $\mathbf{84.61_{0.68}}$ | $\mathbf{64.55_{1.18}}$ | 71.78 |
| all | BERT | ✗ | 82.9 | $\mathbf{69.7}$ | 90.1 | 93.1 | $\mathbf{53.9}$ | 84.8 | 87.7 | 66.1 | $\mathbf{78.54}$ |
| all | Ours | ✓ | $\mathbf{83.6}$ | 69.6 | 90.1 | $\mathbf{93.8}$ | 50.2 | $\mathbf{85.8}$ | 87.7 | $\mathbf{67.3}$ | 78.51 |

Table 2: Results on the GLUE benchmark with K = {200, 500, 1000, all} training samples. Below each task, the total number of training samples for each dataset is indicated. We use F1 for QQP and MRPC, Spearman correlation for STS-B, Matthews correlation for CoLA, and accuracy for the remaining tasks. The standard error is indicated as the subscript.

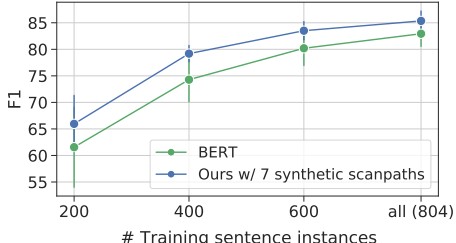

Figure 3: Sentiment classification performance on ETSA in the low-resource setting. Error bars represent the standard error.

thetic scanpaths (the best-performing configuration from the previous experiments). The performance comparison between our model and the baseline model BERT is shown in Figure 3. Our model consistently outperforms BERT, with larger improvements observed when using less training data.

### 3.3 GLUE Benchmark

In contrast to the small and single task-specific ETSA data set, we extended our evaluation to assess whether gaze data could enhance language models across different tasks, including scenarios with substantial text data. To achieve this, we evaluate our model on the GLUE benchmark, a comprehensive collection of 8 diverse NLP tasks with a large number of text samples. As no eye gaze data is available for GLUE, we focus on the comparison with the BERT baseline, and investigate both, high- and low-resource settings.

**High-Resource Performance**  The results of our model on the GLUE test set using all training samples (K = all) are reported in the bottom two rows of Table 2. The results are obtained from the GLUE leaderboard. Our model outperforms BERT in 4

out of 8 tasks, and achieves comparable performance in 3 tasks. However, our model's performance is notably poor in the CoLA task, possibly due to the model's emphasis on gaze sequence ordering, potentially overshadowing the importance of the original word order, which is critical to determine linguistic acceptability of sentences.

**Low-Resource Performance**  We present the results on the GLUE benchmark with K = {200, 500, 1000} training samples in Table 2. We take additional 1,000 samples from the original training set as the development set used for early stopping. The original development set is utilized for testing. We perform 5 runs with different random seeds to shuffle the data and report the average results.

Overall, our model consistently outperforms BERT across tasks, except for the STS-B task. In terms of average score, our model shows performance gains of 2-4% compared to BERT.

## 4 Discussion and Conclusion

We developed a model that integrates synthetic scanpath generation into a scanpath-augmented language model. We observe that the model achieves results that are comparable to a language model augmented with human scanpaths, which eliminates the need for human scanpaths during both training and testing. Human gaze data are only available for a very limited number of NLP tasks and data sets. At application time, under any standard use case scenario of NLP tasks, no gaze recordings are available. Synthetic gaze data not only open the possibility to train high-capacity gaze-augmented models across tasks, which would otherwise require the collection of an impractical large volume of gaze data, but also allow for the

exploitation of eye gaze signals as model input at application time.

Using the GLUE benchmark, we observe that gaze signals show benefits not only for sentiment classification tasks (SST-2), as reported in previous research, but also for entailment classification tasks (MNLI, RTE) and a sentence similarity task (STS-B). This highlights the potential of integrating cognitive signals from eye gaze into a wider range of NLP tasks in the future. Nevertheless, it is evident that not all tasks derive equal benefits from gaze data. It remains up to future research to explore which types of tasks benefit most from gaze signals.

Our results further show that the potential benefit of augmenting language models with gaze data is higher for low-resource settings. Hence, we believe that the augmentation with gaze data might be particularly interesting for low-resource languages. Two ongoing multi-lab efforts to collect large multilingual eye-tracking-while-reading corpora (MECO[4] and MultiplEYE[5]) include a range of low-resource languages, which will allow for training scanpath generators and augmenting language models with synthetic eye gaze for these languages in the near future.

## Limitations

One limitation of our work is that the scanpath generation model Eyettention was pre-trained on eye-tracking data recorded on isolated sentences (*single sentence reading* paradigm). Since the majority of tasks in the GLUE benchmark involve two-sentence classification, future work could involve pre-training the model on an eye-tracking data set specifically designed for two-sentence reading tasks to enhance its performance. Additionally, scanpath augmentation turned out to be detrimental to the language model's performance for the task of identifying linguistically acceptable sentences (CoLA). This finding was to be expected as the actual word order is more relevant for linguistic acceptability of a sentence than the order in which the words are fixated. Pre-training the scanpath generator on an eye-tracking corpus that includes both acceptable and unacceptable sentences may be beneficial for improving the model's performance.

Furthermore, in our proposed framework, the sampling process involved in scanpath generation

---

[4] https://meco-read.com
[5] https://multipleye.eu

during training and at inference time is not conducive to a high model efficiency. Future work could explore alternative scanpath generation models that do not rely on auto-regressive architectures to improve efficiency.

## Ethics Statement

It is crucial to acknowledge potential privacy risks in collecting, sharing, and processing human gaze data. Since eye movements are highly individual, it can be possible to extract a participant's identity from gaze data (Jäger et al., 2020; Makowski et al., 2021). Other personal information such as gender (Sammaknejad et al., 2017) and ethnicity (Blignaut and Wium, 2014) that can be detected to some degree today may turn out to be extractable accurately in the future, which incurs a risk of leakage of personal information from gaze data. Synthetic gaze data can reduce the need for large-scale experiments with human subjects, even though some amount of human gaze data is still necessary to train generative models.

## Acknowledgements

This work was partially funded by the German Federal Ministry of Education and Research under grant 01│S20043.

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

Table 3: Optimal number of scanpaths used for our model in GLUE Benchmark with K = {200, 500, 1000, all} training sentences.

| K | MNLI | QQP | QNLI | SST-2 | CoLA | STS-B | MRPC | RTE |
|---|------|-----|------|-------|------|-------|------|-----|
| 200 | 5 | 3 | 3 | 7 | 3 | 7 | 3 | 7 |
| 500 | 7 | 5 | 3 | 3 | 3 | 7 | 3 | 7 |
| 1000 | 3 | 5 | 5 | 7 | 3 | 5 | 7 | 7 |
| all | 2 | 2 | 4 | 2 | 3 | 3 | 3 | 3 |

# Appendix for Pre-Trained Language Models Augmented with Synthetic Scanpaths for Natural Language Understanding

## A  Model Details

**PLA-AS Framework**  For the PLM-AS framework, we adhere to the design of the original paper (Yang and Hollenstein, 2023). The scanpath encoder consists of a single-direction GRU layer (Cho et al., 2014) with a hidden size of 768 and a dropout rate of 0.1. We initialize the hidden state of the scanpath encoder using the [CLS] token outputs from the final layer of BERT.

## B  Training Details

We train all neural networks using the Py-Torch (Paszke et al., 2019) library on an NVIDIA A100-SXM4-40GB GPU using the NVIDIA CUDA platform. For training, we use the AdamW optimizer (Loshchilov and Hutter, 2019), and a batch size of 32. We train 20 epochs and select the model with the best validation performance for evaluation. The training is early stopped if the validation performance does not increase for 3 consecutive epochs. During the training of our model, we employ the Gumbel-softmax distribution with a temperature hyperparameter set to 0.5. We use the pre-trained checkpoints from the HuggingFace repository (Wolf et al., 2020) for the language model BERT$_{BASE}$.

**Sentiment Classification**  During training, each scanpath associated with one sentence is treated as a separate instance. However, during evaluation, the pre-softmax logits obtained from multiple scanpaths associated with the same sentence are averaged to generate a single prediction for this sentence. We use a learning rate of 1e-5 for training all investigated models.

**GLUE Benchmark**  We evaluate each GLUE data set using the metric specified in the benchmark. We use the code provided in the Hugging-Face repository [6] to train the BERT model and compute the metrics.

In the high-resource setting, we fine-tune the BERT model using the hyperparameter tuning procedure outlined in the original paper (Devlin et al., 2019). We select the best learning rate from {5e-5, 4e-5, 3e-5, 2e-5} for each task based on the performance on the development set. The same learning rate is used for training our model.

Additionally, for our model, we perform a hyperparameter search on the development set to determine the optimal number of scanpaths to be used by the model for each task. We explore different numbers of scanpaths from {2, 3, 4} and select the configuration that achieves the best performance on the development set. The optimal configuration for each task can be found in Table 3.

In the low-resource setting, we use the same learning rate that was found optimal in the high-resource setting for each task. Besides, we perform a hyperparameter search on the development set, investigating different numbers of scanpaths from {3, 5, 7} to be used by our model. The optimal configurations for each task can be found in Table 3.

To reduce variance, we apply shuffling to the training data using 5 different random seeds. We use the first K samples as the new training set, and the subsequent 1,000 samples as the development set. The data seeds used for shuffling are {111,222,333,444,555}, while the seed s=42 is consistently used for model training across all models. The procedure was adapted from Mao et al. (2022).

---

[6] https://github.com/huggingface/transformers/tree/main/examples/pytorch/text-classification