# OpenReview forum: "Pre-Trained Language Models Augmented with Synthetic Scanpaths for Natural Language Understanding"
_EMNLP/2023/Conference — EMNLP 2023 Main_

### Official Review · Reviewer_biQN · 2023-08-03

**Soundness:** 4

**Excitement:**

4: Strong: This paper deepens the understanding of some phenomenon or lowers the barriers to an existing research direction.

**Paper Topic And Main Contributions:**

This paper presents an architecture for integrating synthetic gaze data (generated scanpaths) into a pre-trained language model, combining previous work on generating scanpaths with previous work on augmenting a language model with scanpaths obtained from humans. The model is evaluated on various NLP tasks, and shown to improve on the BERT baseline, especially in low-resource scenarios.

**Questions For The Authors:**

Question A: In l. 264, you say that your model consistently outperforms BERT across tasks, except for STS-B. However, QNLI with K=200 BERT wins with a noticeable difference of almost 3 points. Do you have a hypothesis for the better performance of BERT for QNLI with K=200?

**Reasons To Accept:**

Improvement of the performance of a pre-trained LM through synthetic gaze data is achieved:
* Synthetic gaze data does comes at a cheaper price as gaze data obtained from humans
* Performance improvement on GLUE w.r.t unaugmented BERT is shown to carry over between different NLP tasks, especially in a low-resource scenario with less training data

**Reasons To Reject:**

No reasons to reject.

**Reproducibility:**

4: Could mostly reproduce the results, but there may be some variation because of sample variance or minor variations in their interpretation of the protocol or method.

**Reviewer Confidence:**

3: Pretty sure, but there's a chance I missed something. Although I have a good feel for this area in general, I did not carefully check the paper's details, e.g., the math, experimental design, or novelty.

**Typos Grammar Style And Presentation Improvements:**

* Typo: 295: eyet-racking --> eye-tracking
* I think the accessibility of the "Joint Modeling for NLP tasks" paragraph could be much improved by placing Fig. 1 on page 2, so it can be used as a visual orientation. It would also bring it closer to Sec. 3.2.
* Space allowing, a footnote about the possibility and the potential effects of using models other than BERT_{BASE} (l. 135) could be added.

---

> ### Author Rebuttal · Authors · 2023-08-28
>
> Thank you for your valuable comments. We would like to answer your questions as follows.
> &nbsp;
> &nbsp;
>
> Q1:  In l. 264, you say that your model consistently outperforms BERT across tasks, except for STS-B. However, QNLI with K=200 BERT wins with a noticeable difference of almost 3 points. Do you have a hypothesis for the better performance of BERT for QNLI with K=200?
>
> **A1:** While we may not have a specific hypothesis for this particular task, we can provide some general insights. In our approach, the Eyettention model (used for generating the synthetic scanpaths) is first pre-trained on a natural reading task and subsequently fine-tuned to adapt it to specific NLP tasks. However, in certain cases, such as specific tasks or data domains, the fine-tuning process might be challenging due to limited training data. As a result, the model may not be able to fully adjust and generalize effectively in those cases.
>
> Future research could explore the pre-training (or fine-tuning) of the Eyettention model on task-specific eye movement data rather than natural (i.e., task-unspecific) reading, even with limited data. This approach may result in a better downstream performance for NLP tasks that are similar to the tasks the humans performed during the recording of their eye movements.
> &nbsp;
> &nbsp;
>
>
> Q2: Typos Grammar Style And Presentation Improvements.
> 1. Typo: 295: eyet-racking --> eye-tracking
> 2. I think the accessibility of the "Joint Modeling for NLP tasks" paragraph could be much improved by placing Fig. 1 on page 2, so it can be used as a visual orientation. It would also bring it closer to Sec. 3.2.
> 3. Space allowing, a footnote about the possibility and the potential effects of using models other than BERT_{BASE} (l. 135) could be added.
>
> **A2:** Thank you for pointing this out. We have carefully revised the manuscript according to the reviewer's comments as follows:
> 1. Typo: line 295, *eyet-racking* has been replaced by *eye-tracking*.
> 2. Figure 1 has been positioned on page 2.
> 3. A footnote has been added in line 135 after BERT_BASE: *Note that BERT can be substituted by other state-of-the-art pre-trained large language models from which (subword-) token embeddings can be extracted, potentially leading to further enhancements in task performance.*

---

### Official Review · Reviewer_ER3y · 2023-08-05

**Soundness:** 4

**Excitement:**

3: Ambivalent: It has merits (e.g., it reports state-of-the-art results, the idea is nice), but there are key weaknesses (e.g., it describes incremental work), and it can significantly benefit from another round of revision. However, I won't object to accepting it if my co-reviewers champion it.

**Paper Topic And Main Contributions:**

This paper focuses on utilizing gaze data to improve NLP understanding tasks. The authors propose a unified model that integrates a scanpath generation model (Deng et al., 2023) and the PLM-AS framework (Yang and Hollenstein, 2023), which is a language model augmented with human scanpaths. By synthesizing the gaze data, the proposed model can be applied to the tasks without the annotations of human eye movement, effectively alleviating the data scarcity issue. It is shown that the proposed model performs on par with PLM-AS that takes use of real gaze data on sentiment classification. In addition, the model performs better than the baseline BERT on GLUE benchmark in a low-resource setup.


**Questions For The Authors:**

- Q1: In Yang and Hollenstein (2023), the LPM-AS performance on ETSA (table 1 in their paper) is 90.48 in F1, which differs from the reproduced 85.81. Are these two numbers comparable? If so, what's the reason for this discrepancy?
- Q2: Any insights behind the observation that, in Figure2, the performance peak is when the number of scanpaths is 7 and the performance degrades when considering more scanpaths?
- Q3: In the low-resource setup for GLUE, why not use the original test set for testing?
- Q4: What’s the train/dev/test split in ETSA dataset?


**Reasons To Accept:**

* Novelty of the proposed model - the integration of the scanpath prediction model and the eye-tracking featured model alleviates the data scarcity problem and is shown promising for NLP understanding tasks.
* Paper is well-written, well-structured and motivated.


**Reasons To Reject:**

- One concern is that when the number of scanpath is less than 5 (i.e., 1 and 3 in Figure 2), the proposed model performs worse than the underlying models (i.e., BERT and PLM-AS) and the strong baseline ScanTextGan. This indicates that somehow the model relies on the aggregation of predicted scanpaths. More detailed analysis on this would benefit the paper.
- Other minor concerns are listed in the question list below.


**Reproducibility:**

4: Could mostly reproduce the results, but there may be some variation because of sample variance or minor variations in their interpretation of the protocol or method.

**Reviewer Confidence:**

4: Quite sure. I tried to check the important points carefully. It's unlikely, though conceivable, that I missed something that should affect my ratings.

---

> ### Author Rebuttal · Authors · 2023-08-28
>
> Thank you for providing your insightful feedback. We would like to address your concerns and answer your questions as follows.
> &nbsp;
> &nbsp;
> Q1: One concern is that when the number of scanpath is less than 5 (i.e., 1 and 3 in Figure 2), the proposed model performs worse than the underlying models (i.e., BERT and PLM-AS) and the strong baseline ScanTextGan. This indicates that somehow the model relies on the aggregation of predicted scanpaths. More detailed analysis on this would benefit the paper.
>
> **A1:** We agree that more analysis could enhance the interpretation of these results. In response, we have revised the manuscript and inserted the following paragraph after line 236:
> *However, when using only a small number of scanpaths (1 and 3), our model exhibits lower performance compared to BERT. This could be attributed to the present framework's token embedding rearrangement based on fixation order, resulting in the removal of certain tokens. This process could potentially lead to information loss, thereby impacting the model's performance. This observation underscores the significance of combining a certain quantity of scanpaths, enabling the model to gather information from various perspectives.*
> &nbsp;
> &nbsp;
>
> Q2: In Yang and Hollenstein (2023), the LPM-AS performance on ETSA (table 1 in their paper) is 90.48 in F1, which differs from the reproduced 85.81. Are these two numbers comparable? If so, what's the reason for this discrepancy?
>
> **A2:** These two numbers are not directly comparable due to differences in the evaluation protocols. In the paper by Yang and Hollenstein (2023), their evaluation protocol, as outlined in section 4.2, involves a single train-test split. Since there is no standard train-test split for this dataset, we assume that they have conducted random subset selections for testing, followed by 25 runs with varying random model initializations. In contrast, for our reproduction, we chose a more robust approach: 10-fold cross-validation, aimed at providing a more comprehensive and accurate evaluation of the model's performance. These differing evaluation methodologies explain the observed performance discrepancy.
> &nbsp;
> &nbsp;
>
> Q3: Any insights behind the observation that, in Figure 2, the performance peak is when the number of scanpaths is 7 and the performance degrades when considering more scanpaths?
>
> **A3:**
> 1) The number of scanpaths used by the model is regarded as a hyperparameter in our experiments. To clarify this, we have inserted the following footnote in line 220 after the words 'seven scanpaths':
> *The optimal number of scanpaths to be used by the model is considered a hyperparameter for the subsequent experiments.*
> In Appendix Table 3, when examining the optimal number of scanpaths used for our model across different tasks within the GLUE benchmark, We observe that the optimal number of scanpaths for peak performance is likely linked to the size of the training sentences. In cases where the training sentences are limited in size, a higher number of scanpaths is often more helpful in enhancing the model's generalization performance.
>
> 2) We hypothesis that the observed performance degradation when incorporating more scanpaths could be attributed to a potential bias introduced by excessive scanpath inclusion. This bias may lead the model to overly focus on scanpath patterns, shifting its attention away from refining meaningful token and sentence representations and instead adapting to the variability in scanpaths.
> &nbsp;
> &nbsp;
>
> Q4: In the low-resource setup for GLUE, why not use the original test set for testing?
>
> **A4:** In the low-resource setting, we aimed to reduce variance by conducting 5 runs with distinct random seeds and averaging their results. However, this led to the need to submit 15 full results (one full result including all nine tasks) for the three values of the training data size K which we investigated in the low-resource setting, while the leaderboard allowed only 2 submissions per day. Due to the tight deadline, we decided to prioritize submitting results to the leaderboard for the high-resource setting, ensuring we could meet the evaluation deadline promptly.
> &nbsp;
> &nbsp;
>
> Q5: What’s the train/dev/test split in ETSA dataset?
>
> **A5:** Since the ETSA dataset lacks a standard/predefined train-test split and its size is relatively modest, we opted for a 10-fold cross-validation evaluation approach (line 186).  We split the text into 10 parts, and for each CV fold, we use 9 parts for training and keep one part for testing the model. Additionally, we selected 10\% of the training data as dev data to perform early stopping during training. Our rationale for employing cross-validation stems from its ability to deliver a more comprehensive and accurate evaluation of the model's performance. Mathematically, this approach reduces both bias and variance in the performance estimation, in contrast to the single random train-test split utilized in prior research.

---

### Official Review · Reviewer_3vtZ · 2023-08-10

**Soundness:** 3

**Excitement:**

3: Ambivalent: It has merits (e.g., it reports state-of-the-art results, the idea is nice), but there are key weaknesses (e.g., it describes incremental work), and it can significantly benefit from another round of revision. However, I won't object to accepting it if my co-reviewers champion it.

**Paper Topic And Main Contributions:**

This paper introduces a new method to improve the performance of natural language comprehension, namely using synthesised eye gaze data to augment a pre-trained language model.
The results shows that by adding the synthesised gaze data, the performance of language model is in general improved, which has been tested in sentiment classification task and GLUE benchmark.
According to author(s), gaze data in usual case is hard to get due to its scarce nature and expensive to collect.
This paper thus shows us the potential usage of synthesised gaze data, which cut off the reliance on eye tracking data.

**Reasons To Accept:**

1. The idea of using synthesised eye gaze data to improve language model is novel and interesting.

2. Based on the results, the synthesised gaze data could improve the performance of language model, which could be considered as a nice contribution in terms of short paper based case study.

**Reasons To Reject:**

In Table 2, by increasing the data size K, the advantage of using synthesised gaze data become less obvious.
For example when using all of the training data, the F1 score is basically the same for both of the model(see the Avg column), could this mean that, as long as we have enough language data, it is unnecessary to have gaze data anymore? Gaze data is only valuable when the corresponding language resources are limited? When I check this point with figure 3, I feel the necessity of using larger K size to check if the performance of Bert will be no different from the model with eye gaze.
In a word, further experiment(for the figure 3) should be performed and the conclusion should rather be, under "low resource" situation, synthesised gaze data could help improve the performance of the language model.



**Reproducibility:**

4: Could mostly reproduce the results, but there may be some variation because of sample variance or minor variations in their interpretation of the protocol or method.

**Reviewer Confidence:**

4: Quite sure. I tried to check the important points carefully. It's unlikely, though conceivable, that I missed something that should affect my ratings.

**Typos Grammar Style And Presentation Improvements:**

In Page 3, the part about low resources, the K set should be {200,400, 600, 800} according to Figure 3.

---

> ### Author Rebuttal · Authors · 2023-08-28
>
> Thank you for providing your insightful feedback. We would like to address your concerns as follows.
> &nbsp;
> &nbsp;
>
> Q1: In Table 2, by increasing the data size K, the advantage of using synthesised gaze data become less obvious. For example when using all of the training data, the F1 score is basically the same for both of the model(see the Avg column), could this mean that, as long as we have enough language data, it is unnecessary to have gaze data anymore? Gaze data is only valuable when the corresponding language resources are limited?
>
> **A1:** We agree with the reviewer's observation regarding Table 2, where the advantage of synthesized gaze data becomes less prominent as training data size increases. However, when using all of the training data and examining the model's performance on individual tasks, we still observe performance improvements for specific tasks, namely MNLI, SST-2, STS-B, and RTE, as described in Section 3.3, paragraph ``High-Resource Performance" (line 244). We analyzed these results in Section 4 (line 281), highlighting the potential utility of gaze data for enhancing certain NLP tasks, even when there is an abundance of text data available.
>
> We appreciate the reviewer's feedback, which has prompted us to include a more explicit discussion. While gaze data demonstrates benefits for certain tasks in the high-resource setting, it's evident that not all tasks benefit equally from its incorporation within the current framework.  In response, we have revised our manuscript by inserting the following sentences on line 288:
> *Nevertheless, it's evident that not all tasks derive equal benefits from gaze data. In future research, exploring which types of tasks may benefit most from gaze signals would be of significant interest.*
> &nbsp;
> &nbsp;
>
> Q2: When I check this point with figure 3, I feel the necessity of using larger K size to check if the performance of Bert will be no different from the model with eye gaze. In a word, further experiment(for the figure 3) should be performed and the conclusion should rather be, under "low resource" situation, synthesised gaze data could help improve the performance of the language model.
>
> **A2:** Thank you for bringing this point to our attention. The ETSA sentiment classification dataset comprises only about 1k sentences (line 171). When reserving 10\% for testing, the training data is limited to a maximum of around 800 sentences. Consequently, the rightmost dots in Figure 3 correspond to the scenario where all available training data is used. To clarify this, we have made the following revision in line 228:
> *To test this hypothesis, we sample a small subset of the training sentences, K = {200, 400, 600}, from the total number of approximately 800 training instances.*
> Additionally, in Figure 3, we have updated the label on the x-axis to replace the last number *'800'* with *'all (804)'*.
>
> We would like to underline that the collection of high-quality gaze data for an extensive text corpus is a notably resource intense endeavor (besides the reimbursement of the participants, eye-tracking lab facilities and trained research assistants to supervise participants on a
> one-to-one basis are needed). Consequently, all existing eyetracking datasets for NLP tasks are very limited in size. Overcoming this major bottleneck for the field was a key factor motivating our decision to investigate the potential of synthetic gaze data.
>
> While expanding the size K for further experimentation within the ETSA dataset was not possible, we conducted supplementary experiments using the GLUE dataset in our paper. This provided us with an opportunity to evaluate whether synthetic scanpaths could benefit language models even in scenarios with a large amount of text data.  Notably, our results revealed that for the SST-2 dataset, similar to the ETSA dataset, which also involves sentiment classification, our model augmented with synthetic scanpaths demonstrated improved performance even when utilizing the entirety of the available training data. To clarify this, we have now inserted the following sentences in line 238:
> *In contrast to the small and single-task-specific ETSA dataset, we extended our evaluation to assess whether gaze data could enhance language models across different tasks, including scenarios with substantial amounts of text data. To achieve this, we evaluate our model on the GLUE benchmark.*
> &nbsp;
> &nbsp;
>
>
> Q3: In Page 3, the part about low resources, the K set should be {200,400, 600, 800} according to Figure 3.
>
> **A3:** Please also refer to our answer above concerning the training data size K. We have revised the manuscript in line 228:
> *To test this hypothesis, we sample a small subset of the training sentences, K = \{200, 400, 600\}, from the total number of approximately 800 training instances.* Additionally, in Figure 3, we have updated the label on the x-axis to replace the last number *'800'* with *'all (804)'*. We thank the reviewer for helping us achieve greater clarity.

---

### Meta-Review · Area_Chair_tvXo · 2023-09-12

**Recommendation:** 5

**Metareview:**

This paper explores the question of whether scanpaths (from eye-tracking data) can improve various NLU tasks when the scanpaths used are fully synthetic, based on well known models of human scanpath data. The results show that the answer is yes, though the improvements diminish - unsurprisingly - as the amount of training data increases.

Overall this is a very interesting result, and my opinion as AC is that the authors have addressed the majority of the critical concerns brought up by reviewers. My only additional concern is that this paper does not really fit in "human centered NLP" as the point of this paper is exactly to remove humans from the process, not figuring out how to center humans. So I would suggest moving tracks.

---

### Decision · Program_Chairs · 2023-10-07

**Decision:**

Accept-Main

**Comment:**

This paper explores the question of whether scanpaths (from eye-tracking data) can improve various NLU tasks when the scanpaths used are fully synthetic, based on well known models of human scanpath data. The results show that the answer is yes, though the improvements diminish - unsurprisingly - as the amount of training data increases.

Overall this is a very interesting result, and my opinion as AC is that the authors have addressed the majority of the critical concerns brought up by reviewers. My only additional concern is that this paper does not really fit in "human centered NLP" as the point of this paper is exactly to remove humans from the process, not figuring out how to center humans. So I would suggest moving tracks.